# Pullulan Oxidation in the Presence of Hydrogen Peroxide and *N*-Hydroxyphthalimide

**DOI:** 10.3390/ma15176086

**Published:** 2022-09-02

**Authors:** Gabriela Biliuta, Raluca Ioana Baron, Sergiu Coseri

**Affiliations:** “Petru Poni” Institute of Macromolecular Chemistry, 41A Grigore Ghica Voda Alley, 700487 Iasi, Romania

**Keywords:** pullulan, oxidation, hydrogen peroxide, NHPI, PINO radicals

## Abstract

The C-6 in the maltotriose unit of pullulan was oxidized in an alkaline medium (pH = 10), utilizing a green method that included hydrogen peroxide (H_2_O_2_) as an oxidant and *N*-hydroxyphthalimide (NHPI) as a catalyst for various reaction times. The structure of the resulting oxidized pullulans (PO) was carefully characterized by titration, intrinsic viscosity, FTIR, ^13^C-NMR, and zeta potential. The content of carboxyl groups in PO was dependent on reaction time and varied accordingly. Furthermore, a fast reaction rate was found in the first 2–3 h of the reaction, followed by a decreased rate in the subsequent hours. FTIR and ^13^C-NMR proved that the selective oxidation of the primary alcohol groups of pullulan was achieved. The oxidation also caused the glycoside linkages in the pullulan chain to break, and the viscosity of the pullulan itself went down.

## 1. Introduction

Polysaccharide modification is crucial in the realm of sustainable chemistry. Polysaccharides are excellent starting materials for defined alterations and specialized applications because of their abundance and structural diversity. The conversion of neutral polysaccharides to anionic polymers via polyuronides may result in improved functional characteristics for the polymer [1]. In addition, converting neutral polysaccharides to polyuronides has permitted the manufacture of biomedical nanofibers [2,3]. Several modification procedures try to convert hydroxyl groups from polysaccharides to carboxyl groups, which improves surface polarity and can be used for further surface functionalization. Recently, research has focused on determining the most suitable conditions for polysaccharide functionalization by combining the use of multiple oxidative agents [4]. Polysaccharide oxidation is a recurring problem with numerous applications in a variety of disciplines. To get particular features for polysaccharides or to change existing properties, oxidation is a promising conversion in terms of providing new functional groups, and it is a fairly typical procedure. Despite the fact that oxidation is an important approach for polysaccharide functionalization, it is also one of the most difficult due to the non-selectivity of the reactants and, in particular, material damage during the oxidation process. Finding a technique for effective oxidation of polysaccharides might enhance the quality of the material after oxidation and, indirectly, the subsequent uses. Because of environmental concerns, chlorine-containing oxidants such as NaClO and NaClO_2_ should be avoided whenever possible [5]. As a main oxidant, O_2_, H_2_O_2_, O_3_, or other chlorine-free chemicals are preferred if enough sodium C6-carboxylate groups can be added to oxidized polysaccharides. Among the regularly used methods for the oxidation of polysaccharides, nitroxy radicals are preferred due to their undoubted advantages such as lower costs, friendliness to the environment, high efficiency, low depolymerization, and ease of operation and use. Various methods for oxidation in the presence of free-radical species have been successfully used since the introduction of 2,2,6,6-tetramethyl-pyperidine-1-oxyl radical (TEMPO), sodium bromide, and sodium hypochlorite oxidizing systems [6,7]. A drawback of this method is the inability of the TEMPO reagent to oxidize the polysaccharide without the use of hazardous halogenated oxidizing chemicals, which have the most detrimental impact on the environment. Even if this radical is highly selective, it induces significant depolymerization of the oxidized products. To solve these shortcomings, it was taken into account the use of nonpersistent phthalimide-*N*-oxyl (PINO) radical in situ generated from *N*-hydroxyphthalimide (NHPI) and various cocatalysts able to abstract a hydrogen atom from the O–H bond in NHPI. There are multiple choices for generating PINO radicals, which include metallic (generally transition elements such as cobalt, manganese, lead, copper, iron, nickel, and even their mixtures) [8,9], nonmetallic (peroxides, nitrogen dioxide or nitric acid, anthraquinone, and enzymes of the laccase family) [10,11], and even physical methods (UV/Vis irradiation at wavelength λ = 365 nm) [8]. Any one of these catalysts must be present for the homolytic breakage of the O–H bond in the NHPI to occur, leading to the generation of the PINO radical (Figure 1).

In the field of polysaccharides, the first report on the use of the PINO radical (in situ generated from NHPI) was for cellulose oxidation [12]. Most of these methods of oxidation require an appropriate cocatalyst and the additional presence of sodium hypochlorite and sodium bromide in the system for the PINO radical generation [4,9,12,13,14,15,16]. Other research employed sodium bromide/copper(II) salts [17], anthraquinone/O_2_ [18], or sodium hypochlorite/sodium bromide in a cocatalyst-free system to generate PINO radicals [11,19]. Regardless of the mode of the PINO radical generation, the oxidation takes place selectively at the level of the primary hydroxyl groups of the polysaccharides with the formation of the carboxylic groups. Degumming, the procedure of removing non-cellulosic content from threads such as lignin, hemicellulose, pectin, and other wood elements, is critical to the industrial processing used to separate cellulosic fibers. In an effort to reduce energy consumption, the use of hazardous and expensive solvents, and other factors, researchers are continuously focusing on this crucial process. Due to its high efficiency and energy savings under these circumstances, oxidative degumming has caught the interest of many studies [20]. Under the alkaline degumming with H_2_O_2_ approach, researchers added H_2_O_2_ gradually, attempting to minimize the harm caused by severe H_2_O_2_ oxidation to cellulose in alkaline circumstances [21]. However, H_2_O_2_’s intense oxidation inexorably resulted in the depolymerization of cellulose, harming the fiber. A suggested remedy was the TEMPO system for selective oxidation degumming of ramie [22]. The mechanical qualities of fibers degummed by TEMPO-H_2_O_2_ were superior to those of fibers degummed only by the H_2_O_2_ degumming method because only the primary hydroxyl group at the C6 position of cellulose was selectively oxidized during this process. However, the TEMPO system’s oxidant is NaClO, a chlorine-containing reagent that must be replenished. European and worldwide environmental law is becoming more stringent, recognizing the rising need for halogenated compound substitution at the producing and end-user levels. In a very recent proposed strategy for cellulose degumming, H_2_O_2_ is used in tandem with a catalyst (*N*-hydroxyphthalimide) and a cocatalyst (antraquinone) at pH 10.5 [23]. Under these conditions, the authors claimed a better selectivity for the C6 oxidation and a superior ability of NHPI to capture hydrogen atoms in the primary CH_2_ group, which results in stronger oxidation performance as compared with TEMPO. It might be a little surprising that anthraquinone is present in the suggested system, because it is widely known that H_2_O_2_ may produce the PINO radical from NHPI on its own [8]. In this study, we selected the model polysaccharide pullulan [24,25], which was oxidized with H_2_O_2_ and NHPI at various times in the absence of any cocatalyst, in order to clearly determine the influence of NHPI addition in the case of H_2_O_2_ oxidation of various polysaccharides.

## 2. Materials and Methods

### 2.1. Materials

Pullulan (P) (Mw = 200,000 g/mol) was purchased from Hayashibara Lab. Ltd. (Okoyama, Japan); hydrogen peroxide (H_2_O_2_, 30%, Chimreactiv, Bucuresti, Romania) and *N*-hydroxiphtalimide (NHPI, Fluka, Buchs, Switzerland) were of laboratory grade and used without further purification. Spectrophotometric-grade acetonitrile (CH_3_CN, Sigma Aldrich Co., St. Luis, MO, USA) distilled over CaH_2_ was used as a solvent for NHPI.

### 2.2. H_2_O_2_/NHPI-Mediated Oxidation of Pullulan

The PINO radical was in situ generated from NHPI using H_2_O_2_ as an oxidant. The oxidation of pullulan was carried out under the following conditions: pullulan (0.2 g) was dissolved in a mixture of deionized water (35 mL) and acetonitrile (10 mL). Then, NHPI (1 mM/g pullulan) and H_2_O_2_ (300 mM/g pullulan) were added to the solution, and the pH was adjusted to 10 by the addition of NaOH (2 M) solution. The oxidation reaction was conducted at different reaction times (2 h, 3 h, 5 h, 7 h, and 24 h) at room temperature. A large volume of acetone was used to precipitate the reaction products. The formed precipitate was collected by centrifugation. The recovered precipitate was dissolved in water, and the oligomers were removed by diafiltration through a Millipore ultrafiltration membrane from polyethersulfone (cutoff: 10,000 g·cm^−1^) in an Amicon cell equipped with a tank filled with pure water (conductivity lower than 3 μS/m). When the conductivity of the filtrate was less than 10 mS/m, the diafiltration stopped, and the polymer was recovered by freeze-drying it.

### 2.3. UV Measurements

The electronic absorption spectra were recorded using a SPECORD 200 Analytik Jena spectrometer. UV/Vis recordings were performed for mixtures of NHPI with H_2_O_2_ in acetonitrile/water, with or without the presence of sodium hydroxide.

### 2.4. Carboxyl Content Determination

The degree of substitution (DS) and carboxyl content of the samples were determined by conductometric titration at 22 °C using a conductivity meter CMD 210 (Radiometer, Copenhagen, Denmark), equipped with a CDM 865 cell. Briefly, 20 mL of aqueous polymer solutions (0.1 wt.%) were adjusted to pH 12 using 0.1 N sodium hydroxide and then titrated with aqueous 0.1 N hydrochloric acid (HCl). The degree of substitution (DS) was calculated using the Equation (1):
(1)$$\mathrm{DS}=\frac{162C\left(V_{2}-V_{1}\right)}{w-36C\left(V_{2}-V_{1}\right)}.$$

The carboxyl content was calculated using Equation (2).
(2)$$\mathrm{Carboxyl content}\left(\frac{\mathrm{mmol}}{g}\right)=\frac{c\left(V_{2}-V_{1}\right)}{w},$$
where (*V*_2_ − *V*_1_) represents the volume difference of HCl solution required to neutralize the carboxylic groups (mL), *C* represents the HCl concentration (mol/L), *w* represents the dry sample weight (g), 162 is the molar mass of the anhydrous glucose unit, and 36 is the molar mass difference between the oxidized sample and the anhydrous glucose unit [26].

### 2.5. Viscosimetric Measurements

Viscosimetric measurements were performed at 25 ± 0.01 °C using an Ubbelohde capillary viscometer in combination with an automatic viscosity measurement system (Lauda Instrument, Dusseldorp, Germany). Polymer solutions were prepared in sodium chloride (NaCl 1 M) solution. The solutions were magnetically stirred for 48 h, and viscosity measurements were taken at least twice to ensure that the data were accurate to within +3%. Flow times were obtained with good reproducibility; the errors were less than 1%.

### 2.6. Zeta-Potential (ζ) Measurements

Zeta potential (ζ) was measured using a dynamic light scattering technique (Zetasizer model Nano ZS, Malvern Instruments, Malvern, UK) with a red laser at 633 nm (He/Ne). Zeta potential (ζ) was calculated from the electrophoretic mobility (µ) determined at 25 °C. For kα > 1 (k is the Debye–Huckel parameter, and α is the particle radius), the Smoluchowski relationship was used (Equation (3)).
(3)$$\zeta =\frac{\eta µ}{\varepsilon },$$
where η is the viscosity, and ε is the dielectric constant.

### 2.7. FTIR Measurements

Infrared absorption spectra of pullulan and oxidized pullulan samples were recorded using a Bruker Vertex 70 spectrometer in a scan range from 4000 cm^−1^ to 650 cm^−1^, at a resolution of 2 cm^−1^ over 32 scans. Samples were measured as KBr pellets.

### 2.8. NMR Analyses

Samples were prepared for ^13^C NMR by dissolving 50 mg in 0.7 mL of D_2_O. The obtained solution was filtered through a pipette containing glass wool into a standard 5 mm NMR tube. The NMR spectra were taken with a Bruker Advance DRX400 MHz Spectrometer that had a 5 mm QNP direct detection probe and z-gradients.

## 3. Results

### 3.1. UV Measurements

In order to verify if the generation of the PINO radical really takes place in the conditions designed by us, we found it of interest to study the in situ generation of the PINO radical from NHPI in the presence of H_2_O_2_ using UV/Vis absorption spectroscopy. The UV/Vis experiments (Figure 2a) show that the NHPI exhibited strong absorption at 294 nm. When H_2_O_2_ was added to NHPI, a new absorption band centered at 430 nm replaced the maximum absorption band of NHPI. This is clear evidence of the formation of PINO radicals, which could then act as a mediator for pullulan oxidation. In addition, it was observed that the PINO radical was generated only when the pH of the H_2_O_2_ solution was adjusted to 10 with NaOH solution (2M). When the pH was lower (below 10), the specific absorption of the PINO radical (430 nm) was not observed (Figure 2b). Using UV/Vis, we were able to prove the formation of the PINO radical in the system. This evidence can serve to further understand NHPI’s role in the oxidation reaction mechanism.

### 3.2. The Oxidation of Pullulan in the Presence of the H_2_O_2_/NHPI System

Compared to the traditional oxidation process, H_2_O_2_ oxidation is more effective and less harmful to the environment [27,28]. H_2_O_2_ can be activated in alkaline media but is highly stable in an acidic environment [29]. In alkaline surroundings, H_2_O_2_ is decomposed into several species including free radicals such as O^−2•^, OH^•^, OOH^•^, and OH^−^, which exhibit significant oxidizing properties. However, they also virtually harm the polysaccharide substrate, which may have an impact on future application of the oxidation products. As is well known, H_2_O_2_ is not a selective oxidant for polysaccharides such as cellulose [30]. All three OH groups in the cellulose structural unit can be oxidized into CHO, C=O, and COOH. On the one hand, hydrogen peroxide has the capacity to activate the cellulose glycosidic bonds, resulting in the breakdown of macromolecules during oxidation. Previous research has shown that the rate at which H_2_O_2_ decomposes has a significant impact on that compound’s capacity to oxidize, and the pH level influences that rate [31,32,33]. We believed that further research on a water-soluble polysaccharide, such as pullulan, in terms of oxidation with NHPI and H_2_O_2_ would provide new information on this system. In this way, we could bring new evidence in support of all the abovementioned observations in corroboration with those made in a very recent paper in which NHPI was used in combination with H_2_O_2_, as well as in the presence of a cocatalyst (anthraquinone), for the degumming process [23].

Therefore, we used NHPI as a catalyst and H_2_O_2_ in alkaline conditions (pH = 10), in order to ensure the reaction’s selectivity and avoid any cocatalyst (Figure 3). By promoting the oxidation of a variety of organic compounds, NHPI demonstrated its extraordinarily potent catalytic action. Its ability to abstract a hydrogen atom through a homolytic scission of the >N–O–H bond, forming the active species, phthalimide-*N*-oxy radical (PINO), is well documented [8]. In order to selectively oxidize the primary hydroxyl groups of the C6 position in pullulan to carboxilic groups, the oxidizing agent (H_2_O_2_) activates the PINO radical to generate oxoammonium cations [34]. In this process, the PINO radicals might have two concurrent tasks: the first is to promote the oxidation process, and the second is to inhibit the oxidation process due to the capture of free radicals (originating from the H_2_O_2_).

Table 1 shows the amount of introduced carboxyl groups and the yields of oxidized pullulan (PO). In order to assess the potency of the H_2_O_2_/NHPI system to oxidize the pullulan, a sample of pullulan was oxidized under the same conditions in terms of time (5 h), pH, and amount of H_2_O_2_, but in the absence of the NHPI catalytic agent. Comparing the data, it is worth noting that the time of the reaction has a great impact on the oxidation reaction. It was observed that the amount of introduced carboxylic groups is strongly dependent on the reaction time. Thus, at the first stages, when the oxidant concentration was high, there was a sharp increase in the carboxyl group formation. Apparently, prolonging the reaction time (more than 5 h) did not cause any further increase in the amount of produced carboxylic groups; conversely, the amount tended to steadily decrease. It is worth mentioning that this drop in the carboxyl group concentration was caused by the intense depolymerization phenomena associated with the pullulan chain, favored by the formed COOH (or even aldehyde as an intermediate oxidized species) groups, which act as catalytically active sites for the scission of neighboring β-1,4-glycosidic linkages [35]. Thus, the depolymerized low-molecular-weight species were removed during the dialysis process, which led to an apparent decrease in the COOH groups on the larger oxidized fractions [36,37]. The drop in H_2_O_2_ concentration in the system was also a result of the presence of NHPI, likewise playing the role of trapping the potential free radicals generated by H_2_O_2_, which can be utilized to explain the decrease over time in the content of carboxylic groups attached to the pullulan. The PINO radical, also produced by H_2_O_2_ consumption, was responsible for this uptake (radical species derived from H_2_O_2_ absorption of proton from >N–O–H). Additionally, the presence of radical species in the system significantly contributed to the depolymerization of the polymer chain, which was facilitated (catalyzed) by the aldehyde groups generated. This depolymerization eventually resulted in low-molecular-weight compounds, which could not be isolated by precipitation, explained by the ongoing decline in the yield in oxidized products (Table 1). When pullulan was oxidized employing only H_2_O_2_ (in the absence of NHPI) under the same conditions, a larger amount of introduced carboxyl groups was found (2.17 mmol COO^−^/g pullulan, PO-5 h*). In the same condition of oxidation (300 mM H_2_O_2_, 5 h), the oxidation with H_2_O_2_ was non-selective; any of the OH groups could be oxidized [27,28]. On the basis of these findings, we concluded that the PINO radical had an inhibitory effect on the oxidation process after 5 h of reaction, as the COO^−^ content began to decrease slightly (0.67 mmolCOO^−^/g pullulan for PO-7h; 0.63 mmol COO^−^/g pullulan for PO-24 h).

The carboxyl content, degree of substitution, and yield all depend on the reaction duration, as shown in Figure 4. As indicated in the graph, all of these components had maximum values for a reaction time of up to 3 h. As the reaction time approached 5 h, it becomes apparent that it did not vary considerably, maintaining a rather constant state. 

For the evaluation of the intrinsic viscosity of oxidized pullulan, an aqueous solution of NaCl (1 M) was used. Thus, the oxidized pullulan samples behaved like a neutral polymer, with the polyelectrolyte effect being suppressed and the intrinsic viscosity reflecting only the coil dimensions. After oxidation of the pullulan in the presence of H_2_O_2_/NHPI, it was observed that the intrinsic viscosity of the oxidized pullulan was smaller than the initial pullulan. When comparing the viscosity data with the DS obtained for the oxidized pullulan samples, it was observed that the intrinsic viscosity remained approximately constant with the increase of the DS (Figure 5). By comparing the values obtained for intrinsic viscosity, we can appreciate that the pullulan chain degradation appeared at a very early stage, with the viscosity decreasing from 0.776 dL/g (P) to 0.123 dL/g (PO-2h); therefore, the chain breakdown occurred simultaneously with oxidation. After 2 h of reaction, the intrinsic viscosity was reduced by approximately 84%, suggesting a fast and severe degradation of macromolecules in the reaction medium.

Under these conditions, a natural and inescapable question arises. Is the degradation caused by a larger degree of substitution, a longer response time, or both of these factors working together? To find a solution to this issue, we need to first make an effort to alter only one of the parameters that might lead to such a significant breakdown of the polymeric chains. A study is currently being conducted in order to establish the chain degradation that occurs during the oxidation reaction. The study is looking at what happens to the same sample (with a given degree of oxidation) when it is exposed to the reaction medium for different amounts of time during the oxidation reaction. This strategy could help us learn more about the competition between oxidation and degradation events that happens during the H_2_O_2_-mediated oxidation of pullulan when it is used with long reaction times.

The zeta potential of all oxidized samples was measured in deionized water at a concentration of 1 mg/mL. The carboxyl content of oxidized pullulan had a great impact on the zeta potential. Higher absolute values of zeta potential resulted in improved suspension stabilities for negatively charged surfaces, causing more electrostatic repulsion between nanoscale particles. The zeta potential values for pullulan and oxidized pullulan are shown in Figure 6. The resulting values of zeta potential were strongly dependent on the number of negatively charged groups introduced into the polymeric chain after oxidation, ranging from a ζ of −19 ± 0.2 mV (1.33 mmol COO^−^/g pullulan) in the sample oxidized for 2 h (PO-2h) to a ζ of −2.35 ± 0.6 mV (0.63 mmol COO^−^/g pullulan) in the sample oxidized for 24 h (PO-24h). It is clear that oxidation occurred smoothly in the first 2–3 h of the reaction. An increase in reaction time inevitably led to a decrease in zeta potential, due to the degradation of the oxidized products and their conversion into low-molecular-weight compounds.

### 3.3. FTIR Analysis

FTIR spectroscopy is the most commonly used method for determining pullulan’s structural features. By displaying stretching and bending vibrational peaks at their respective wavenumbers, the presence of different sorts of interconnections and connections in pullulan can be established using the KBr pellet technique [38,39]. Pullulan shows the typical FTIR spectrum of polysaccharides (Figure 7a). The glucopyranosyl rings of pullulan have α-(1-4) and α-(1-6) linkages in their structure. To generate maltotriose repeating units, three glucose moieties are joined by α-(1-4) connections, and each maltotriose is linked by α-(1-6) linkages. The peaks at 756 cm^−1^ and 930 cm^−1^, respectively, indicated the presence of α-(1-4) and α-(1-6) bonds in the pullulan structure [40]. The large and strong absorption band at 3436 cm^−1^ indicated the OH stretching vibrations in the pullulan structure [41]. The other strong absorption at 2924 cm^−1^ indicated an *sp*^3^ C–H stretching vibration [42]. The peak at 1381 cm^−1^ confirmed the C–O–H bending at the C-6 position [43].

The peak at 1159 cm^−1^ in pullulan suggested C–O–C bond stretching vibrations and the existence of a glyosidic bridge [44]. The large peak at roughly 1020 cm^−1^ can be attributed to the stretching vibration of C–O at the C-4 position of a glucose molecule in the pullulan structure [45]. Due to the interaction of CH_2_, C1–H, and C5–H bending vibrations, the distinctive band at about 849 cm^−1^ signifies the α-glucose configuration [46].

The FTIR spectra of oxidized pullulan at different reaction times of oxidation are shown in Figure 7b. The IR spectra of oxidized samples show some noteworthy changes. As a result of the conversion of OH groups to COO^−^ groups, the band at 3436 shifted from 3417 cm^−1^ (PO-2h) to 3401 cm^−1^ (PO-24h). The oxidized materials’ spectra showed a new band, which was not observed in the pullulan, around 1665 cm^−1^, i.e., 1663 (PO-2h), 1665 (PO-3h), 1665 (PO-5h), 1666 (PO-7h), and 1665 (PO-24h), corresponding to stretching vibration C=O bonds, a band that indicates pullulan oxidation [47,48]. Furthermore, all curves showed absorption peaks at 1153 cm^−1^, attributed to the asymmetric stretch vibration of the bridge bond C–O–C. The band intensity decreased in PO-24h. This reduction in C–O–C stretching indicates the severe breakage of the glycosidic bond. In conclusion, the FTIR measurements show that the H_2_O_2_/NHPI system successfully oxidized the original pullulan with a drastic degradation in the case of oxidized pullulan, after 24 h (PO-24h).

### 3.4. NMR Analysis

The chemical structure of both pullulan and oxidized pullulan was also studied using ^13^C-NMR spectroscopy. According to the literature [7,49], when the hydroxyl groups on C-6 of polysaccharides are converted to carboxyl groups by the TEMPO oxidant system, the spectra of ^13^C-NMR for oxidized polysaccharides reveal a characteristic absorption peak of around 170 ppm. ^13^C-NMR for oxidized pullulan (PO) revealed that selective oxidation of the primary alcohol groups could be achieved by the H_2_O_2_/NHPI system. The changes in ^13^C-NMR of P and PO with different carboxyl content are shown in Figure 8 and Figure 9. From the pullulan ^13^C-NMR spectra, the characteristic peaks for pullulans were as follows: 100.19 (C1-4g), 97.89 (C1-6g), 77.33 (C4), 70.29–73.41 (C2,3,5), 66.44 (C6-6g), and 60.36 (C6-4g) ppm (Figure 8) [50]. In the oxidized pullulan (Figure 9), a new peak appeared at 175 ppm, with the peak intensity increasing with the increase of the carboxyl content (PO-2h had the most intense peak, while PO-24h had the lowest peak). This new peak shows that carboxyl groups were added to pullulan at the C6 position.

## 4. Conclusions

In this study, the ability of an oxidation system based on NHPI and H_2_O_2_ in the absence of any cocatalyst that yields the PINO radical was tested using the water-soluble polysaccharide pullulan. However, H_2_O_2_ was still involved in the production of the PINO radical from NHPI, as shown through UV/Vis measurements. The content of the carboxylic groups introduced in the pullulan backbone depended significantly on the reaction time and pH value, which was adjusted to 10. The oxidized pullulan samples were characterized by FTIR and ^13^C-NMR techniques. A potential issue when H_2_O_2_ was employed for the oxidation was represented by the degradation of the polymer chains in the first hours of reaction. Therefore, the yield of oxidized pullulan decreased while the time of oxidation increased. Decreasing the reaction time and the amount of H_2_O_2_ could lead to a decrease in the degradation of the macromolecules.

## Figures and Tables

**Figure 1 materials-15-06086-f001:**
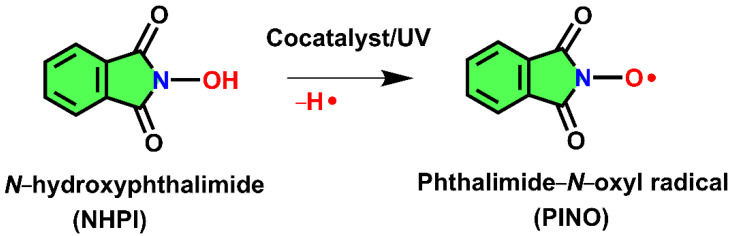
The schematization of the generation of the PINO radical.

**Figure 2 materials-15-06086-f002:**
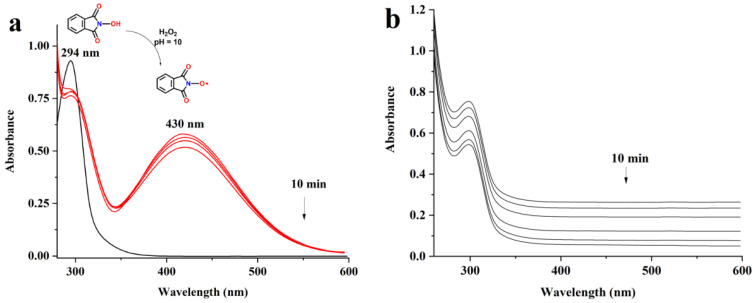
(**a**) UV/Vis absorption of NHPI (1 mM) and H_2_O_2_ (100 mM), in acetonitrile/water 1:1 (vol) at pH 10. (**b**) UV/Vis absorption of a mixture of NHPI (1 mM) and H_2_O_2_ (100 mM) in acetonitrile/water 1:1 (vol), without pH correction to 10.

**Figure 3 materials-15-06086-f003:**
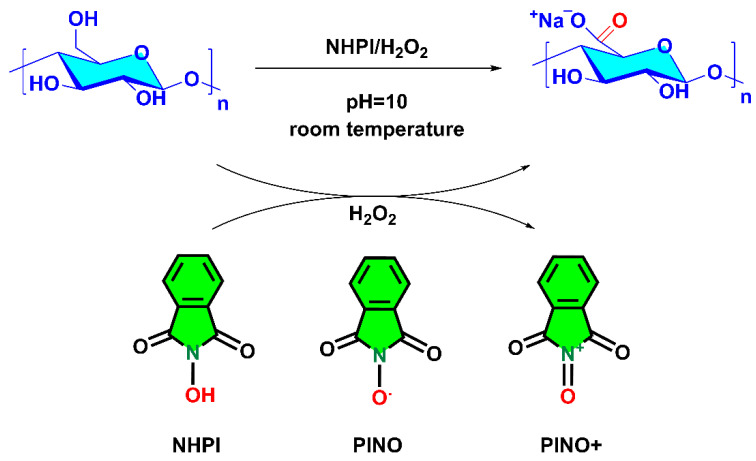
Pullulan oxidation scheme mediated by NHPI/H_2_O_2_ at pH = 10.

**Figure 4 materials-15-06086-f004:**
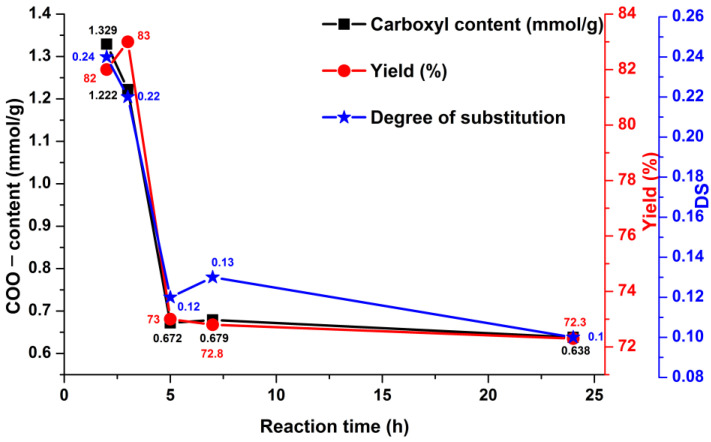
The effect of reaction time on the carboxyl content, degree of substitution, and yield of the oxidized pullulan.

**Figure 5 materials-15-06086-f005:**
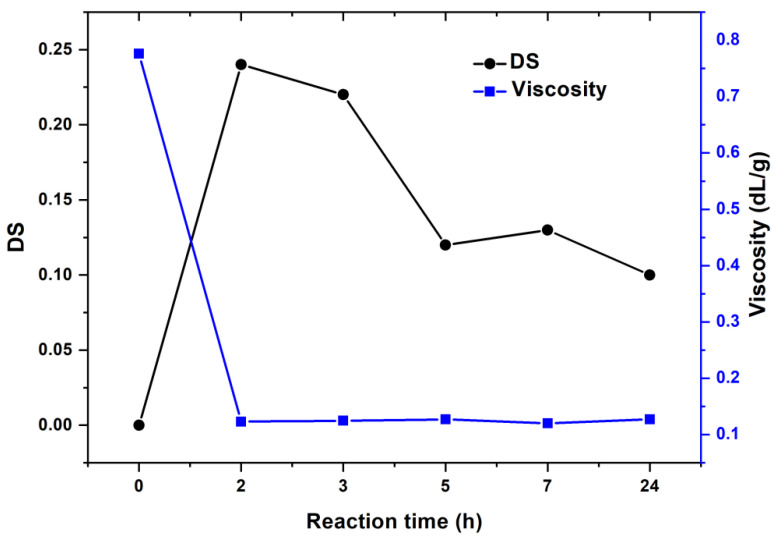
Effect of the intrinsic viscosity on the degree of substitution (DS).

**Figure 6 materials-15-06086-f006:**
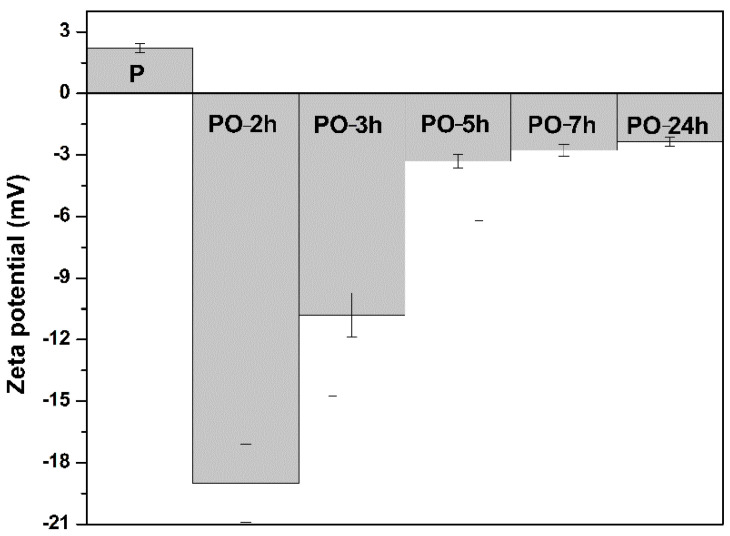
The zeta potential of pullulan (P) and oxidized pullulan (PO) at different reaction times.

**Figure 7 materials-15-06086-f007:**
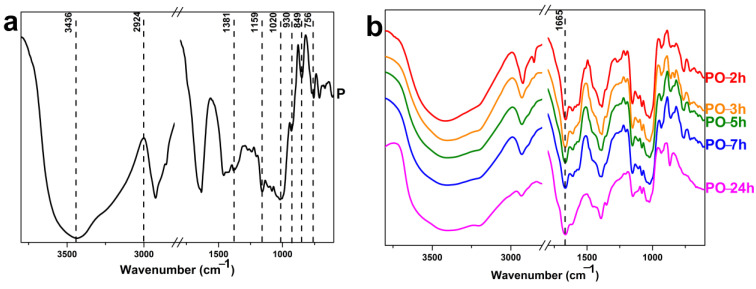
(**a**) FTIR spectrum of pullulan (P). (**b**) FTIR spectra of oxidized pullulan (PO) in the presence of an H_2_O_2_/NHPI system.

**Figure 8 materials-15-06086-f008:**
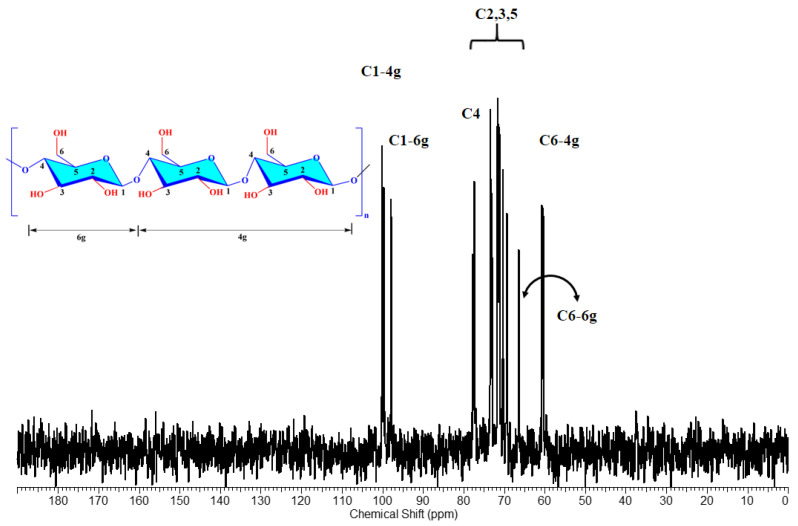
^13^C-NMR spectrum of pullulan (P).

**Figure 9 materials-15-06086-f009:**
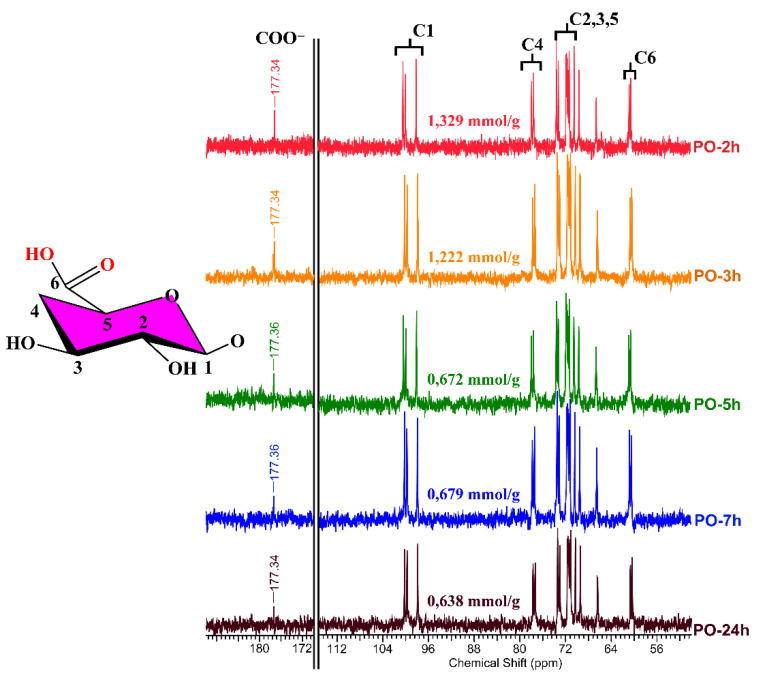
^13^C-NMR spectra of oxidized pullulan in the presence of H_2_O_2_/NHPI system (PO—oxidized pullulan at different reaction times).

**Table 1 materials-15-06086-t001:** Amount of negatively charged groups (COO^−^), reaction time, and yield for oxidized pullulan (PO), at different reaction times and pH = 10. * Oxidation in the presence of H_2_O_2_, without NHPI in the same condition.

Run	Reaction Time (h)	Amount of Negatively Charged Groups COO^−^ (mmol/g)	Mass Yield (%)
PO-2 h	2	1.33	83
PO-3 h	3	1.22	82
PO-5 h	5	0.70	77
PO-5 h *	5	2.17	70
PO-7 h	7	0.67	73
PO-24 h	24	0.63	72

## Data Availability

Not applicable.

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
