# Peer review of "Pullulan Oxidation in the Presence of Hydrogen Peroxide and N-Hydroxyphthalimide"

_materials, 2022, doi:10.3390/ma15176086_

Round 1
Reviewer 1 Report
The work described in the manuscript adds some insights in the selective C-6 oxidation of polysaccharides. Although not a breakthrough, the work is provided of some interest and should be published. Nevertheless, prior to be considered for publication, the manuscript needs some revision regarding the English language and some inaccuracies, e.g.:
- In line 54 the authors refer to a hydrogen atom as a proton;
- In line 73 the authors refer to lignin, hemicellulose and pectin as gums;
- In line 86 the authors refer to the hypochlorite ion as chloride ion (and say this is harmful for the environment!);
- In line 128 a C is missing in ºC;
- In equations 1 and 2 a dot or a space should be used for multiplications, instead of an x, which can be misleading; also a parenthesis is missing in equation 1, and the authors call (V2-V1) a volume, when it is a volume difference.
Author Response
Dear Editor,
Dear Reviewers,
Firstly, we found the reviewers' comments very pertinent and useful for us; we thank them for the valuable comments and suggestions. We carefully revised the manuscript according to their recommendations. Below are the point-to-point changes we made in our revised manuscript.
Response to Reviewer 1:
Thank you for your time and for your considerations on the manuscript; they have been helpful in improving our work. All the changes made in the manuscript were highlighted in red.
- In line 54 the authors refer to a hydrogen atom as a proton;
The revised manuscript has now changed as follows:
To solve these shortcomings, it was taken into account the use of non-persistent phthalimide-N-oxyl (PINO) radical in situ generated from N-hydroxyphthalimide (NHPI) and various cocatalysts able to abstract a hydrogen atom from the O-H bond in NHPI.
- In line 73 the authors refer to lignin, hemicellulose and pectin as gums;
The revised manuscript has now changed; we have reformulated the entire paragraph to avoid any misunderstanding:
Degumming, the procedure of removing non-cellulosic content from threads such as lignin, hemicellulose, pectin, and other wood elements, is critical to the industrial processing used to separate cellulosic fibers.
- In line 86 the authors refer to the hypochlorite ion as chloride ion (and say this is harmful for the environment!);
Thank you for your observations. In the revised manuscript, the sentence was rewritten accordingly.
However, the TEMPO system's oxidant is NaClO, a chlorine-containing reagent that must be replenished. European and worldwide environmental law is becoming more stringent, recognizing the rising need for halogenated compound substitution at the producing and end-user levels.
-In line 128 a C is missing in ºC;
Thank you for your observations. It was corrected.
- In equations 1 and 2 a dot or a space should be used for multiplications, instead of an x, which can be misleading; also a parenthesis is missing in equation 1, and the authors call (V2-V1) a volume, when it is a volume difference.
Thank you for the suggestion. It was adjusted.

Reviewer 2 Report
In my opinion, the work is out of the scope of the journal Materials. The authors have submitted the manuscript to section “Catalytic Materials” for the special Issue “Advances in Nanostructured Catalysts”. The work does not fit into these topics because they have used a molecular/homogeneous catalytic system to perform the oxidation of pullulan and not a nanostructured/material catalyst.
Furthermore, the authors focus on the oxidation of the expensive pullulan as model polysaccharide, which is not a topic of real interest. The main potential interest of the pullulan lies in its applications in the food, pharmaceutical, and biomedical fields; and the search for more cost-effective ways to produce pullulan.
The authors could have used another model polysaccharide or explored any of the applications of the oxidized pullulan they produced.
The authors did not cite references on the same topic they have published recently, as for example:
- doi:10.3390/ma12213585
- http://dx.doi.org/10.1016/j.eurpolymj.2016.10.020
And some important references on the modification of pullulan, such as:
- http://dx.doi.org/10.1016/j.carbpol.2015.01.032
- DOI: 10.1007/s00253-011-3477-y
Due to these observations, I would not recommend that the article be published in the journal Materials.
Author Response
Dear Editor,
Dear Reviewers,
Firstly, we found the reviewers' comments very pertinent and useful for us; we thank them for the valuable comments and suggestions. We carefully revised the manuscript according to their recommendations. Below are the point-to-point changes we made in our revised manuscript.
Response to Reviewer 2:
Thank you for your time and for your considerations on the manuscript; they have been helpful in improving our work. All the changes made in the manuscript were highlighted in red.
- In my opinion, the work is out of the scope of the journal Materials. The authors have submitted the manuscript to section “Catalytic Materials” for the special Issue “Advances in Nanostructured Catalysts”. The work does not fit into these topics because they have used a molecular/homogeneous catalytic system to perform the oxidation of pullulan and not a nanostructured/material catalyst.
Thank you for your observations.
Even if the catalytic system used in this manuscript is not strictly speaking a nano-sized catalyst, the proposed catalytic system fits well under the broad umbrella of the "Catalytic Materials" section, as the Editor of this journal, as well as the reviewer 3 ("I am sure this nice paper will fit in Materials, Advances in Nanostructured Catalysts special issue") emphasized. This manuscript was submitted to this journal based on an invitation for this section.
- Furthermore, the authors focus on the oxidation of the expensive pullulan as model polysaccharide, which is not a topic of real interest. The main potential interest of the pullulan lies in its applications in the food, pharmaceutical, and biomedical fields; and the search for more cost-effective ways to produce pullulan. The authors could have used another model polysaccharide or explored any of the applications of the oxidized pullulan they produced.
Due to these observations, I would not recommend that the article be published in the journal Materials.
Even though pullulan is a more expensive polysaccharide (but the price keeps going down), this type of biopolymer was chosen because it is well known that pullulan, a water-soluble polymer, has special properties that make it useful in a wide range of medical and pharmaceutical applications, in the development of systems for the controlled release of drugs, and in systems that use biocompatible and biodegradable polymers.
It should be noted that its well-known structure (matrotiose units linked by α-(1-4) glycosidic bonds, which are in turn connected by α-(1-6) glycosidic bonds) with primary and secondary hydroxyl groups available for specific reactions, as well as its excellent solubility in water combined with its exceptional film-forming properties, make pullulan a polysaccharide of the moment.
Also, for various applications in the structure of pullulan, the presence of the other groups, apart from the hydroxyl ones, is necessary. These groups can be introduced through different methods of functionalization, later serving as bridges for binding other compounds or precursors in processes of the click-chemistry type. Adding carboxylic groups is a method that could be used for this purpose. This is done through different pullulan oxidation protocols.
All of these things led us to study this new catalytic system on pullulan, the polysaccharide that made it easy to separate the oxidized products, which would not have been possible if we had used cellulose, chitosan, or even starch.
- The authors did not cite references on the same topic they have published recently, as for example:
doi:10.3390/ma12213585
http://dx.doi.org/10.1016/j.eurpolymj.2016.10.020
http://dx.doi.org/10.1016/j.carbpol.2015.01.032
DOI: 10.1007/s00253-011-3477-y
We introduced the reference recommended by the reviewer. Thank you for your suggestion.

Reviewer 3 Report
The systematic modification of natural materials is a key question in the preparation of smart polymers. The manuscript ‘Pullulan oxidation in the presence of hydrogen peroxide and N-hydroxyphthalimide’ submitted by Gabriela Biliuta and co-workers reports the ‘selective’ oxidation of Pullulan as a selected polysaccharide. The authors oxidized Pullulan with ‘in situ’ generated "PINO" radicals (H2O2 and NHPI) in water at pH=10. The resulting mixtures were characterized by C=O content titration, intrinsic viscosity, zeta potential measurements, IR and 13C NMR spectroscopy as well. Unfortunately, the polysaccharide chains have been fragmented during the oxidation processes.
My comments:
1./ Although the manuscript mentioned, it is unclear why the COO- content of the mixtures decreases with time. Please explain this in the manuscript. (line 228-231.)
2./ What ‘oligomers’ are formed during the oxidation? Have these been compared with those that resulted in the presence of the other Oxidizing agents yet?
3./ A suggestion, just: The oxidation of polysaccharides was carried out in the water. Did the authors think about Oxone®, as a green compound (Potassium peroxymonosulfate)?
Overall, this research is good continuous work by her group and after the authors address these short corrections, I am sure this nice paper will fit in Materials, Advances in Nanostructured Catalysts special issue.
Author Response
Dear Editor,
Dear Reviewers,
Firstly, we found the reviewers' comments very pertinent and useful for us; we thank them for the valuable comments and suggestions. We carefully revised the manuscript according to their recommendations. Below are the point-to-point changes we made in our revised manuscript.
Response to Reviewer 3:
Thank you for your time and for your considerations on the manuscript; they have been helpful in improving our work. All the changes made in the manuscript were highlighted in red.
- Although the manuscript mentioned, it is unclear why the COO- content of the mixtures decreases with time. Please explain this in the manuscript. (line 228-231.)
With the increase of reaction time, the content of the carboxylic groups increases at first, and then starts to decrease slightly. At the beginning of the oxidation process (2-3 h), the oxidant concentration is higher and the oxidation reaction occurs smoothly. Increasing the reaction time leads to oxidant consumption, resulting in decreasing reaction efficiency (after 5 h). Therefore, the prolonged reaction time will cause the chain breaking of the pullulan, thus predominating its depolymerized forms. Additionally, it has been noted that the polysaccharide chains' carboxylic groups serve as catalytically active sites for the scission of neighboring β-1,4-glycosidic linkages. (L. Zhou, X. Yang, J. Xu, M. Shi, F. Wang, C. Chen, J. Xu Depolymerization of cellulose to glucose by oxidation–hydrolysis Green Chem., 17 (2015), pp. 1519-1524; G. Rånby, R.H. Marchessault Inductive effects in the hydrolysis of cellulose chains J. Polym. Sci., 36 (1959), pp. 561-564). These fragments with a relatively small molecular weight, below 10,000 Da, will be eliminated in the dialysis purification process (a Millipore ultrafiltration membrane from polyethersulfone, cut-off: 10,000 g cm-1 was used for purification).
For certain purposes in which a charge with high COO¯ is desired, it is obvious that short-time reactions will be used to avoid depolymerization processes.
According to reviewer’s comment, we pointed out this aspect in the revised manuscript:
It has been observed that the amount of introduced carboxylic groups is strongly dependent on the reaction time. Thus, at the first stages, when the oxidant concentration is high, there is a sharp increase in the carboxyl group formation. Apparently, peculiar, prolonging the reaction time (more than 5 h) did not cause any further increase in the amount of produced carboxylic groups, conversely the amount tended to steadily decrease. It is worth mentioning that this drop in the carboxyl group concentration is caused by the intense depolymerization phenomena associated with the pullulan chain, which are favored by the formed COOH (or even aldehyde as an intermediate oxidized species) groups, which act as catalytically active sites for the scission of neighboring beta-1,4-glycosidic linkages. Thus, the depolymerized low molecular weight species are removed during the dialysis process, which leads to an apparent decrease of the COOH groups on the larger oxidized fractions. (L. Zhou, X. Yang, J. Xu, M. Shi, F. Wang, C. Chen, J. Xu Depolymerization of cellulose to glucose by oxidation–hydrolysis Green Chem., 17 (2015), pp. 1519-1524; G. Rånby, R.H. Marchessault Inductive effects in the hydrolysis of cellulose chains J. Polym. Sci., 36 (1959), pp. 561-564).
- What ‘oligomers’ are formed during the oxidation? Have these been compared with those that resulted in the presence of the other Oxidizing agents yet?
During the oxidation, the oxidant attacks the glyosidic bonds and decomposes the macromolecular chain of pullulan. In particular, compared with other oxidation processes in which H2O2 was employed to fabricate the oxidized polysaccharides with high carboxyl content (Cellulose 26, 2699-2713, 2019), during the oxidation the polymer suffers degradation and decomposes into oligomers and soluble organic compounds, such as formic acid, acetic acid, propanoic acid butyl ester, etc.
So, when we refer to oligomers, we refer to short-chain oligomers (fragments with a relatively low molecular weight, below 10,000 Da). Future research will focus on quantifying and identifying these oligomeric fractions produced during the oxidation process.
- A suggestion, just: The oxidation of polysaccharides was carried out in the water. Did the authors think about Oxone®, as a green compound (Potassium peroxymonosulfate)?
Thank you very much for the suggestion and the great idea! We are happy to keep the proposal in mind since we are certain that such a reagent or system of reagents would address many of the issues.

Round 2
Reviewer 2 Report
I believe that the authors adequately answered all questions and amended the manuscript with the suggestions. So, I am now convinced that the manuscript can be accepted for publication.